

# A global behavioural model of human fire use and management: WHAM! v1.0

Oliver Perkins[1,2], Matt Kasoar,[1,3], Apostolos Voulgarakis[1,3,4], Cathy Smith[1,5], Jay Mistry[1,5], James D.A. Millington[1,2]

[1] The Leverhulme Centre for Wildfires, Environment, and Society, Imperial College London, SB7 2BX, UK
[2] Department of Geography, King's College London, WC2B 4BG, UK
[3] Department of Physics, Imperial College London
[4] Atmospheric Environment and Climate Change Laboratory, Technical University of Crete, Kounoupidiana, 73100, Greece
[5] Department of Geography, Royal Holloway, University of London, TW20 0EX, UK

*Correspondence to*: Ol Perkins (oliver.perkins@kcl.ac.uk)

**Abstract.** Fire is an integral ecosystem process and a major natural source of vegetation disturbance globally. Yet at the same time, humans use and manage fire in diverse ways and for a huge range of purposes. Therefore, it is perhaps unsurprising that a central finding of the first Fire Model Intercomparison Project was simplistic representation of humans is a substantial shortcoming in the fire modules of dynamic global vegetation models (DGVMs). In response to this challenge, we present a novel, global geospatial model that seeks to capture the diversity of human-fire interactions. Empirically-grounded with a global database of anthropogenic fire impacts, WHAM! (the **W**ildfire **H**uman **A**gency **M**odel) represents the underlying behavioural and land system drivers of human approaches to fire management and their impact on fire regimes. WHAM! is designed to be coupled with DGVMs (JULES-INFERNO in the current instance), such that human and biophysical drivers of fire on Earth, and their interactions, can be captured in process-based models for the first time. Initial outputs from WHAM! presented here are in line with previous evidence suggesting managed anthropogenic fire use is decreasing globally, and point to land use intensification as the underlying reason for this phenomenon.



## 1. Introduction

Fire is a fundamental earth-system process and a key driver of global vegetation dynamics (Pausas et al., 2017). Yet at the same time humans use fire for a large range of purposes (Smith et al., 2022), from disposal of agricultural residues (Lin and Begho, 2022), to social ceremonies (Beaulaton, 2010). Anthropogenic fire management strategies are similarly diverse, spanning preventative strategies such as indigenous patch burning (Laris, 2002) to fire exclusion through industrial fire extinguishing (Eloy et al., 2019). Furthermore, humans not only influence fire regimes directly, through starting and managing fires, but also indirectly, by altering and fragmenting fuel-loads (Harrison et al., 2021; Rosan et al., 2022), for example through road-building (Haas et al., 2022), livestock grazing (Archibald, 2016) and logging (Cochrane and Barber, 2009).

As such, present-day wildfire regimes are best understood as a coupled socio-ecological system (Kelley et al., 2019; Ford et al., 2021), in which people are the largest driver of changes to the frequency, intensity and extent of fire (Kelly et al., 2020; Andela et al., 2017). Given the extent and diversity of human-fire interactions, it is perhaps not surprising that the recently completed Fire Model Intercomparison project (FIREMIP; Hantson et al., 2016) found representation of humans is a substantial shortcoming in current fire-enabled dynamic global vegetation models (DGVMs). Representation of humans was the biggest cause of disparity between model outputs (Teckentrup et al., 2019) and a major contributor to divergence between models and remotely sensed observations (Forkel et al., 2019). Models did not agree on the magnitude nor the direction of the anthropogenic influence on burned area over the last century (Teckentrup et al., 2019).

Underlying this issue are DGVMs' simplistic representations of human activity. To this point, inclusion of anthropogenic influences on fire regimes have been limited to globally homogenous functions based on population density and / or GDP (Rabin et al., 2018; Ford et al., 2021). These approaches treat all anthropogenic fires as similar events, and therefore do not account for the diverse ways in which humans use and manage fire in contrasting land use systems and underlying socio-ecological contexts. This makes separating the role of biophysical drivers of fire regimes from human fire use and management, and from anthropogenic changes to fuel load, a substantial research challenge (Jones et al., 2022). Indeed, integration of managed anthropogenic fire into models at all scales has been identified as a major step required to 'reimagine fire science for the Anthropocene' (Shuman et al., 2022). Lack of adequate representation of humans in DVGMs limits their ability to predict the future of fire on Earth, which, consequently, affects the quality of forecasts of future emissions of carbon and air pollutants (Lasslop et al., 2019) used in Earth system models (ESMs) used to predict climate change.

Here, we present a new model that seeks to address this challenge. Drawing on agent-based approaches, the model – *WHAM!* (*W*ildfire *H*uman *A*gency *M*odel) – captures the drivers and distribution of human fire use and management globally. Importantly, it represents the influence of *categorical* differences in land use systems – arable farming, livestock farming, forestry, and non-extractive land uses such as conservation and recreation – on human fire management strategies.



One reason for currently limited approaches to representing anthropogenic impacts on fire regimes has been a lack of
systematic data from which to derive alternative parameterisations (Forkel et al., 2019; Jones et al., 2022). WHAM!'s empirical
foundation is the Database of Anthropogenic Fire Impacts (DAFI), which was developed to address this issue (Millington et
al., 2022). DAFI contains data from 1809 case studies of human-fire interactions, sourced primarily from the academic
literature, but also the 'grey' literature of government and NGO reports. DAFI, which is freely available online (Perkins and
Millington, 2021), enables WHAM! to represent the processes that drive human impacts on fire regimes from the bottom up.

A concurrent data issue has been that the majority of anthropogenic fires have not been captured in global-scale Earth
observation products (Zhang et al., 2018). Anthropogenic fires are typically small: >50% are smaller than the 21ha size at
which MODIS can reliably detect them (Millington et al., 2022). This has made the evaluation of representations of
anthropogenic fire challenging, often leading to circular calibration of modelled fire counts and/or burned area to a structurally
biased observational record (Teckentrup et al., 2019). However, with recent advances in fine-scale remote sensing of burned
area (e.g. Gaveau et al., 2021; Chen et al., 2023), it is now possible to capture much more of the anthropogenic signal. As such,
with the combination of DAFI and fine-scale Earth observation products, an independent evaluation of process-based model
representation of anthropogenic influences on fire regimes is now possible.

Anthropogenic fire can be broadly categorised into three components: managed fire, unmanaged fires and escaped fires
(UNEP, 2023). For managed fire, Millington et al., (2022) identify seven central modes of anthropogenic fire use, which
include ranges from field preparation in shifting cultivation systems to prescribed fire for biodiversity conservation.
Unmanaged anthropogenic fire comprises accidental fires from cigarette butts or machinery failure, as well as arson (Scott,
1985). Escaped fire is when a managed fire grows beyond its original purpose to become an unmanaged wildfire (e.g. Cano-
Crespo et al., 2015). Of these three categories, calculating burned area from managed anthropogenic fire can be done within
*WHAM!* itself, as a function of the land system and land user objectives. However, the burned area from unmanaged and
escaped anthropogenic fires can only be calculated through coupling with a biophysical fire model.

Therefore, as fire regimes emerge from a combination of anthropogenic and biophysical influences, WHAM! has been
developed to be coupled with DGVMs, in the first instance JULES-INFERNO, the fire-enabled dynamic global vegetation
model in the UK Earth System Model (Mangeon et al., 2016; Burton et al., 2019). This model coupling will allow process-
based representations of anthropogenic and biophysical drivers to be integrated to form a cohesive socio-ecological simulation
of global fire regimes. In this paper we present WHAM! in standalone form, and so the model evaluation focuses on managed
fire, and particularly a comparison of cropland fires from WHAM! with the GFED5 crop fires product (Hall et al., 2023). As
WHAM! is an empirical model, the performance metrics of statistical parameterisations against DAFI data are also provided
(Supplement C). However, outside of croplands global burned area products do not differentiate between managed and



unmanaged fires (Chen et al., 2023), so full evaluation of WHAM! will only be possible after integration with JULES-INFERNO, when a complete picture of global burned area can be calculated.

To facilitate model integration, the parameterisation of WHAM! presented in the main text takes relevant biophysical input variables from JULES model outputs. However, as our intention is ultimately that WHAM! can be coupled with multiple fire-enabled DGVMs, we have also parameterised WHAM! using Earth observation products for its biophysical inputs. The differences between the EO-driven version, named 'WHAM-EO' (for WHAM! Earth Observation), and the default version of WHAM! are described in Supplement A.


### 2.    Methods

The typical timesteps adopted by DGVMs (e.g hourly or daily) are not relevant for large-scale modelling of human decision-making (Arneth et al., 2014). As such, WHAM! runs at an annual timestep, inline with other geospatial land use models run at large spatial extents (e.g. Murray-Rust et al., 2014). WHAM! can be parameterised at different spatial resolutions, but is

here set up to run at the spatial resolution of JULES adopted in CMIP6 (1.875° x 1.25°; Wiltshire et al., 2020).

From a model structure perspective, WHAM! replaces globally-uniform functions generating numbers of anthropogenic 'ignitions' with a process-based representation of anthropogenic fire use and management. An overview of these changes is given in Figure 1. WHAM! outputs, therefore, are burned area from managed anthropogenic fire as a fraction of each model

grid cell, unmanaged anthropogenic fires as number of fires $km^{-2}$ $yr^{-1}$, and fire suppression intensity on a dimensionless scale (0-1; Table 1).

WHAM! is presented in the following stages. Firstly, we present the procedure to allocate categorical types of land user spatially (building on Perkins et al, (2022), Section 2.1). Secondly, we describe calculation of anthropogenic managed fire

(Section 2.2), and thirdly, unmanaged fires (Section 2.3). Fire suppression is described in Section 2.4. Setup of historical runs, including model evaluation is described in Section 2.5. WHAM! is written in Python 3.8 using the Agentpy library (Foramitti, 2021). Model code, including forcing data, is made freely available online (Perkins et al., 2023a).



**Table 1: Overview of WHAM! outputs and respective units; burned area from unmanaged anthropogenic fires – including the impact of fire suppression – will be calculated by a DGVM (initially JULES-INFERNO) as a part of a coupled model ensemble.**

| Variable | Section | Output units |
|---|---|---|
| Managed fire | 2.2 | Burned area (fraction of grid cell) |
| Unmanaged fire | 2.3 | Fire counts ($km^{-2}$ $year^{-1}$) |
| Fire suppression | 2.4 | Suppression intensity (0-1; dimensionless) |


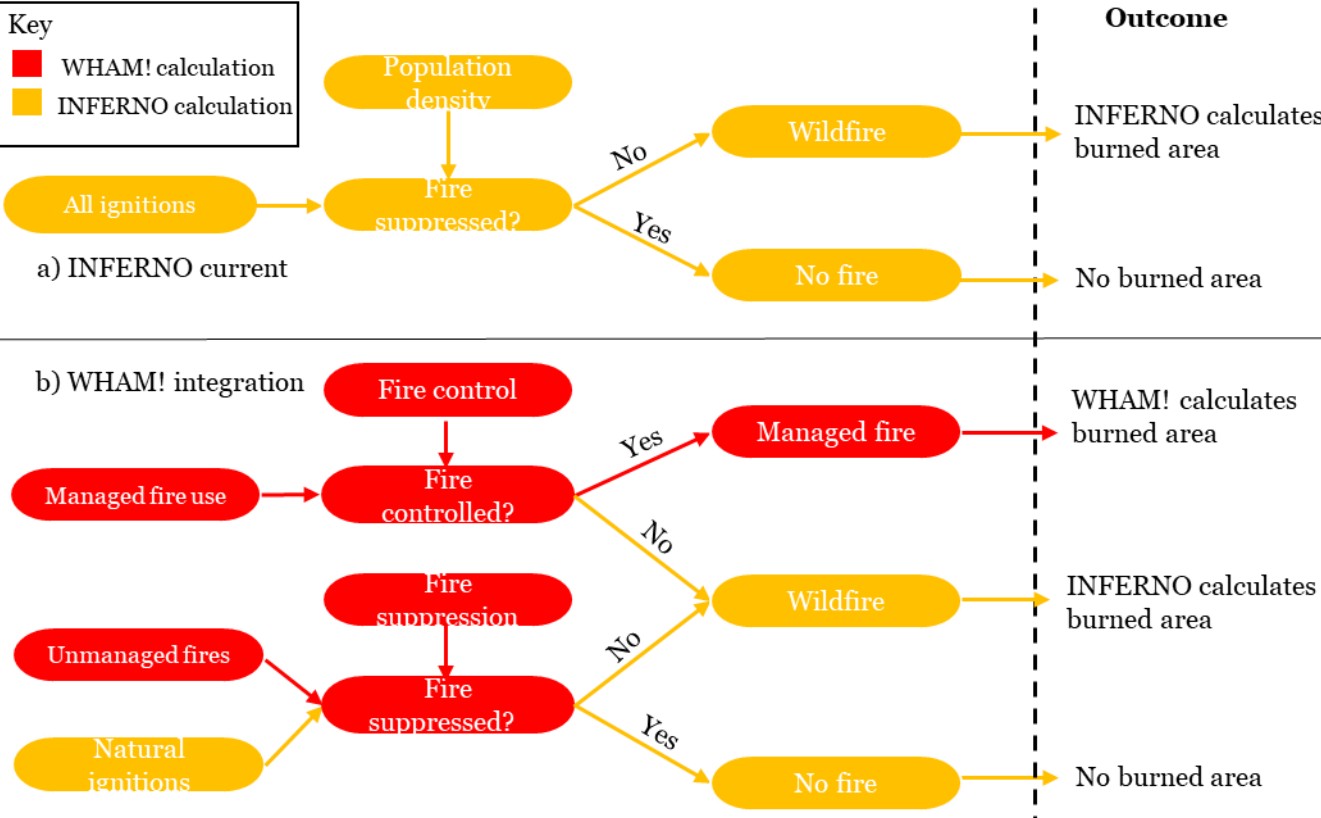

**Figure 1: Schematic representation of the structural changes to INFERNO enabled by WHAM! integration. Rather than treating all fires as similar events, the WHAM! integration can differentiate between managed fires – that spread primarily according to anthropogenic land management objectives – and unmanaged fires, that spread principally according to biophysical drivers.**






## 2.1 Land use in WHAM!

### 2.1.1 Defining agent functional types

WHAM! is driven by the spatiotemporal distribution and decisions of agent functional types (AFTs; Rounsevell et al., 2012). AFTs are analogous to plant functional types (PFTs) in DGVMs, in that they provide broad categories of function or roles that allow global heterogeneity to be represented in a manageable way (Arneth et al., 2014). The AFTs used are derived from a combination of land use system (LUS) types – cropland, livestock farming, forestry and non-extractive uses – and from what we term anthropogenic fire regimes (AFRs). The AFRs are categorical – 'pre-industrial', 'transitional', 'industrial' and 'post-industrial' - and reflect available resources and management perspectives at a given time and place. The underlying theoretical justification and quantitative signatures of these AFRs are presented in Millington et al., (2022). The combination of four land use systems and four anthropogenic fire regimes provides 16 combined land-fire systems (LFS) which in turn can be split into distinct AFTs. In 12 of 16 cases, LFS and AFT are synonymous (Table 2). In the remaining four cases, multiple AFTs compete for space within a single LFS (see Section 2.1.3).

**Table 2: Agent-Functional Types (AFTs, *italicised*) and Land Fire Systems (LFS) resulting from the cross-referencing of Land Use Systems and Anthropogenic Fire Regimes (AFRs). In 12/16 cases, AFTs are synonymous with –LFS, while in the remaining cases the relationship is multi-faceted (and therefore multiple AFTs exist within a LFS as shown by *italics*)**

| *AFR* | *Land Use System* | | | |
|---|---|---|---|---|
| | **Non-extractive** | **Forestry** | **Livestock** | **Cropland** |
| **Pre-Industrial** | Unoccupied | Hunter-Gatherer | Pastoralist | Swidden |
| **Transition** | *Recreationalist, Conservationist* | *Logging, Agroforestry* | Extensive Livestock Farmer | *Small-holder (Subsistence), Small-holder (Market)* |
| **Industrial** | State land manager | Managed Forestry | Intensive Livestock Farmer | Intensive Farmer |
| **Post-Industrial** | *Conservationist, Recreationalist* | Abandoned forest plantation | Abandoned pasture | Abandoned cropland |



### 2.1.2 Land system distribution

To ensure compatibility under model intercomparison project protocols, WHAM! takes landcover inputs from the LUH2
forcing data sets of Hurtt et al., (2020). Cropland, pasture, rangeland and urban land cover fractions were taken directly as
forcing data. However, to calculate the proportion of tree cover used for forestry versus non-extractive land use, as well as the
unoccupied fraction of cell, a process of competition was simulated. This used the same methods as those for the distribution
of AFTs, which is described below in section 2.1.3

### 2.1.3 Agent functional type distribution

The global spatiotemporal distribution of AFTs is based on a simulation of their competition for land. After Arneth et al.,
(2014), we first define the socio-ecological niche of each AFT, before comparing their relative competitiveness in a pixel to
allocate space. A detailed presentation and evaluation of the representation of competition for land in WHAM! is given in
Perkins et al., (2022). Here, we therefore provide a brief summary; a schematic representation is given in Figure 2.

Firstly, to capture the socio-ecological niche of AFTs a simple classification tree was constructed quantitatively for each. This
was done using data in DAFI as the target variable and explanatory variables from the secondary data sets given in Table 3.
Bootstrapping was used to identify the most robust single-tree structure across data sub-samples. The bootstrapping approach
also led to multiple possible split threshold values and output probabilities for the selected structure. These were retained to
express data uncertainty, and to create transitions between output probability spaces ('niches') in the resulting maps. To
prioritise representation of process and to avoid overfitting, the median and modal number of nodes or splits in AFTs' trees is
two, and tree structures were accepted only if they had strong land system process rationale (Perkins et al., 2022). The complete
set of tree models used for AFT distribution is provided along with model code (Perkins et al., 2023b).

Secondly, having defined a single tree per AFT, a process of competition was simulated by normalising the output probabilities
of tree models across relevant AFTs:

$$AFT_{ij} = p(AFT_{ij}) \Big/ \sum p\big(AFT_j[p(AFT_j) > \theta]\big) \tag{1}$$

where $AFT_{ij}$ is the fractional coverage of the i[th] AFT in the j[th] cell, and $p(AFTij)$ and $\sum p(AFTj)$ are the probability of the
classification tree for the i[th] AFT and for all AFTs respectively. Because of the choice of simple tree structures, to avoid very
small land fractions continuing to be allocated to inappropriate AFTs – for example shifting cultivation in the USA Corn belt
– output probabilities beneath a threshold parameter θ were set to 0. In this way, the output probabilities of the tree models of
the AFTs' niche were in effect interpreted as a 'competitiveness score' in a given pixel.



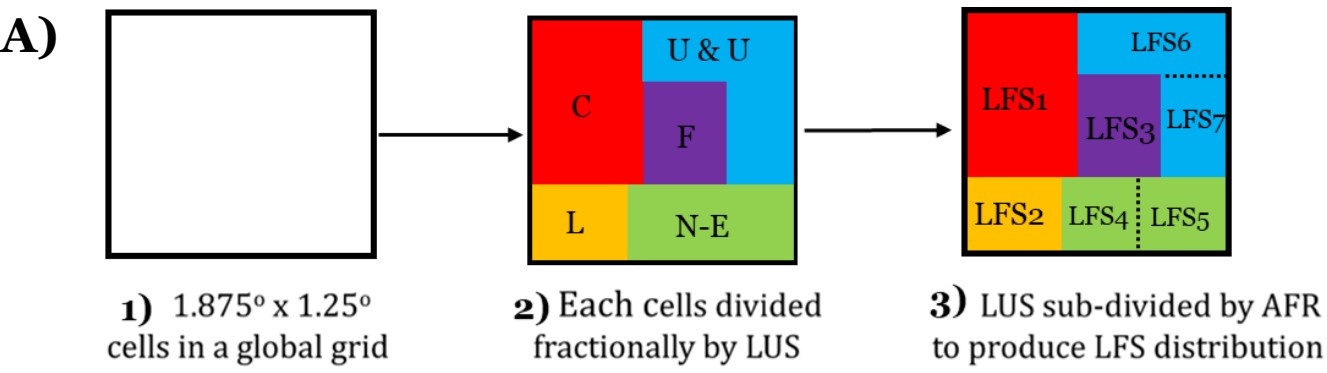

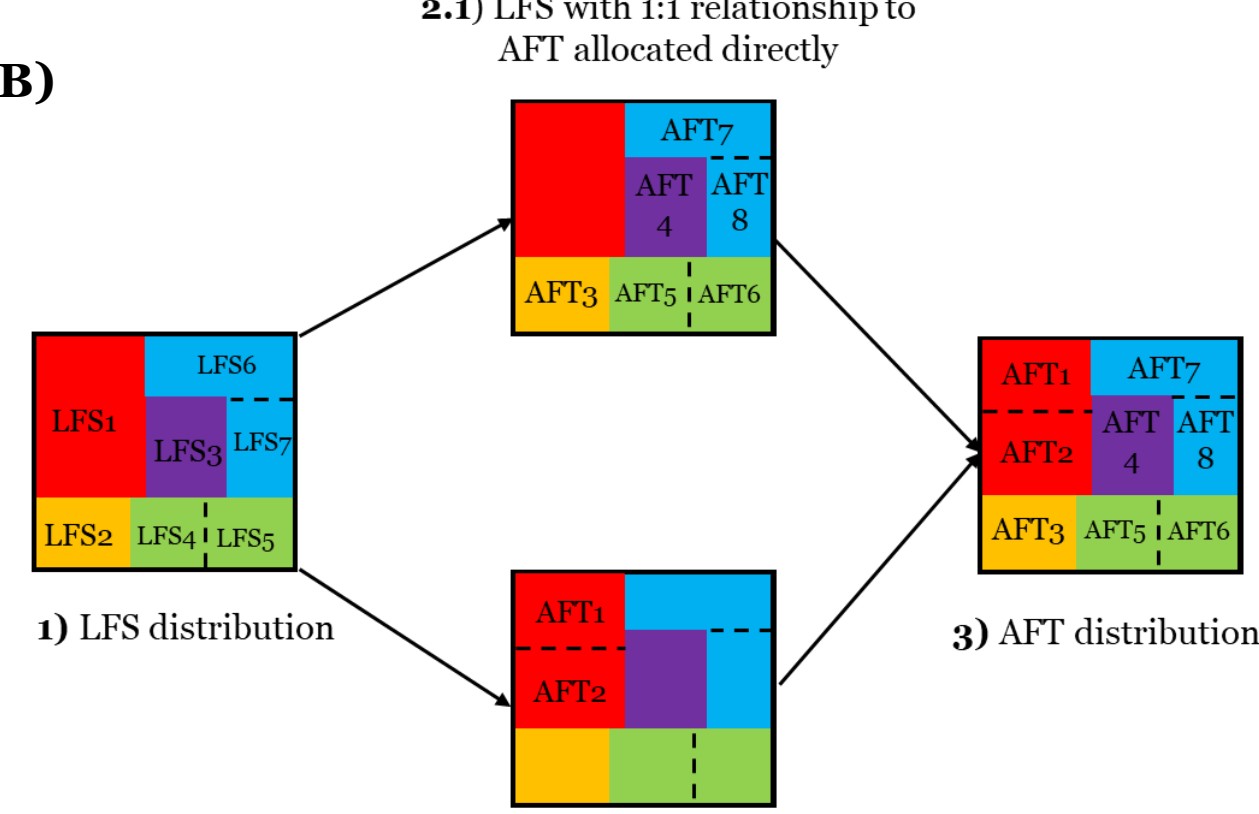

**Figure 2: Overview of agent functional type (AFT) distribution in WHAM!; A) describes how cells are divided first into Land use systems (LUS) and then into land-fire systems (LFS). B) then describes the relationship of land fire systems to AFTs. Key: C = cropland, L = livestock, N-E = non-extractive, F = forestry, U&U = Urban and unoccupied.**




**Table 3: Data sets used to parameterise DAFI submodels for land use competition and agent functional type allocation, managed fire use, unmanaged fires and fire suppression. All data were resampled to the resolution of JULES-INFERNO (1.875º x 1.25º). All data sets have an annual temporal resolution.**

| Variable type | Variable name | Spatial resolution | Temporal range | Source |
|---|---|---|---|---|
| Socio economic | Population density | 0.04º | 2000-2020 | CIESIN, 2017 |
| | Gross Domestic Product | 0.08º | 1990-2015 | Kummu et al., 2018 |
| | Human Development Index | 0.08º | 1990-2015 | Kummu et al., 2018 |
| | Market access[+] | 0.08º | 2000 (1990-2015) | Verburg et al., 2011) |
| Land use & Land cover | Fractional land cover (anthropogenic) | 0.25º | 1990-2020 | Hurtt et al., 2020 |
| | Land cover composition (natural)* | 1.875º x 1.25º | 1990-2020 | Clark et al., 2011 |
| Biophysical | Potential evapotranspiration* | 1.875º x 1.25º | 1990-2014 | Best et al., 2011 |
| | Ecosystem net primary production* | 1.875º x 1.25º | 1990-2014 | Clark et al., 2011 |
| | Topography | 30m | N/A | Van Zyl et al., 2001 |


**Key: + single year of data extrapolated to other years from other secondary data (see Perkins et al., 2022 Supplementary material A). *replaced with Earth observation data in WHAM_EO (Supplement A).**





## 2.2 Managed anthropogenic fire

Analysis of DAFI revealed seven central modes of global anthropogenic fire use (Millington et al., 2022): crop field
preparation; crop residue burning; pasture management; hunting and gathering; pyrome management; vegetation clearance;
and arson. For the first five of these fire uses, a common method of parameterisation was adopted (Section 2.2.1); however,
vegetation clearance (Section 2.2.2) and arson (Section 2.3.2) required bespoke approaches. We define arson as fire use as a
weapon or to cause deliberate property damage (Scott, 1985); therefore it is classed as unmanaged – though intentional – fire
use.


### 2.2.1 Default fire use parameterisation

Fire use for crop field preparation, crop residue burning, pasture management, hunting and gathering and pyrome management
were parameterised similarly. Each AFT was parameterised for both their tendency to practice each of these five modes of fire
use (measured as a probability; hereafter 'fire use tendency') and, where relevant, the extent of their use (measured as burned
area fraction 0-1; hereafter 'fire use extent'). To calculate the burned area fraction for these five modes of fire use for each
AFT, a global map of fire use tendency was multiplied by a map of fire use extent.

Fire use tendency and fire use extent maps were calculated with a combination of statistical tools: classification and regression
trees, generalised linear models (GLMs), and their combination. These tools were chosen for their simplicity, interpretability,
and complementarity. As with the distribution of AFTs, they were constructed using data from DAFI. Owing to data sparsity
and sampling biases, corrections were needed for some combinations of AFTs and fire use mode. For example,
parameterisation of hunter-gatherer fire did not capture the categorical difference between such fire uses in open savannas
versus forest ecosystems. Data in DAFI suggest hunter-gatherer fire in grasslands and savannas burns 18.0% of the land cover
on average compared to 6.7% in forests (Perkins and Millington 2021). Hence a correction was applied to capture this effect.
These fire purpose specific amendments are detailed in Supplement B. A complete set of AFT parameterisations is provided
along with model code, and their evaluation is described in Supplement C.





In addition to extra parameters required for specific fire use modes and AFTs, two global biases in DAFI data were corrected using top-down constraints on managed fire use. These were a vegetation constraint, and a dominant anthropogenic fire regime
(AFR) effect. The vegetation constraint corrected for the lack of DAFI case studies in deserts and other very arid environments (Perkins et al., 2022). This parameterisation is analogous to the use of the fraction of absorbed photosynthetically active radiation (FAPAR) as a vegetation constraint in the LPJ-GUESS SIMFIRE model (Knorr et al., 2014). Similarly, the dominant-AFR constraint was needed as DAFI under-sampled places where fire use was absent in more developed contexts (Perkins et al., 2022). From a process perspective, it aimed to capture the impact of imitation in fire management amongst land users
(Lopes et al., 2020; Cammelli et al., 2020), and also the impact of legal and other social barriers that prevent or restrict managed fire use where fire suppression has become the dominant management paradigm (Kreuter et al., 2008; Harr et al., 2014; Bendel et al., 2020). The vegetation constraint was calculated as:

$$VC_t = \begin{Bmatrix} 1 \ if \ soil_t \leq T_{soil} \\ 1 - soil_t \ otherwise \end{Bmatrix} \tag{2}$$

$\quad BA_t = \ \widehat{BA}_t * VC_t \tag{3}$

where $soil_t$ is the bare soil fraction from JULES outputs at time = t; $T_{soil}$ is a free parameter determining at what fractional coverage of bare soil in a cell the vegetation constraint should apply; $VC_t$ is the vegetation constraint, and $\widehat{BA}_t$ and $BA_t$ are raw burned area from bottom-up AFT calculations, and burned area adjusted for the vegetation constraint.
Similarly, the dominant AFR constraint was applied in model cells where the intensive AFR had the largest coverage of the four AFRs in that cell. It was calculated as:

$$AFRC_t = \begin{Bmatrix} 1 \ if \ Industrial_t \leq T_{AFR} \\ 1 - Industrial_t \ otherwise \end{Bmatrix} \tag{4}$$

$$BA_t = \ \widehat{BA}_t * AFRC_t \tag{5}$$


where $Industrial_t$ is the fractional coverage of the Industrial AFR at time = t; $T_{AFR}$ is a free parameter determining at what fractional coverage the constraint should apply; and $AFRC_t$ is the industrial AFR constraint. As a result of this process, the model gained two free parameters: the two critical thresholds at which the bare soil and dominant AFR constraints take effect. The sensitivity of model outputs to WHAM!'s free parameters is explored in Supplement D.




### 2.2.2 Vegetation clearance

Parameterisation of fire for clearance of primary vegetation (e.g. 'deforestation fire') was complicated by the fact that WHAM! takes land cover inputs from LUH2 (Hurtt et al., 2020). Therefore, rather than seeking to model change in land cover directly, WHAM! instead uses the vegetation transitions specified LUH2 data (Ma et al., 2020). Using these pre-defined land cover
changes between simulated time steps, WHAM! calculates the portion of newly cleared primary vegetation occupied by each anthropogenic fire regime, and on this basis calculates the fraction of the cleared land area that would have involved fire use. WHAM! uses AFRs rather than AFTs for this calculation to ensure robust data samples.

Further, given that it is frequently a clandestine process, vegetation clearance fire proved highly difficult to quantify in DAFI.
Remote sensing data are widely available for the size of cleared patches due to deforestation (e.g. Morton et al., 2006), but not for the specific amount of deforestation driven by differing actors and its relationship to fire. Consequently, DAFI contains 136 measurements of vegetation clearance fire size, but only 20 of burned area (Perkins and Millington, 2021). We therefore parameterise the ratio of area of vegetation cleared to burned area for each AFR as free parameters; given the inherent resulting uncertainty, their impact on burned area is explored in model sensitivity analysis (Supplement D). Initial values for these fire
to deforestation ratios are given in Table 4. The ratio between fire and deforestation was assumed to be 1 for the pre-industrial AFR, as by definition this AFR does not use machinery for land management (Millington et al., 2022). Furthermore, none of the AFTs for the post-industrial AFR would clear primary vegetation for extractive land use systems so there is no ratio for these AFTs.

**Table 4: Ratio of burned area to total area of vegetation cleared used to parameterise vegetation clearance fire use. A ratio of 1.00 means 100% of vegetation was cleared by fire use.**

| Anthropogenic fire regime | Ratio | Source |
|---|---|---|
| Pre-industrial | 1.00 | Ontological: the pre-industrial AFR does not make use of machinery |
| Transitional | 0.84 | Aragão et al., 2008 |
| Industrial | 0.31 | van Marle et al., 2017 |
| Post-industrial | N/A | No post-industrial AFTs cleared vegetation for extractive purposes |



### 2.3 Unmanaged anthropogenic fire

#### 2.3.1 Escaped fires

Escaped fires are those managed anthropogenic fires that escape control measures and grow to become unmanaged wildfires. As with managed fire use, escaped fire parameterisations were derived from data in DAFI. The starting point was the calculation of a baseline escape rate for the six managed fire types described in Section 2.2 (Millington et al., 2022; Table 5). This was then adjusted for the degree of fire control measures applied by an AFT. DAFI represents the degree of control measures applied during managed fire use on a 0-3 ordinal scale. There was a clear divide in outcome between no or little

control (i.e. 0 or 1) and moderate or intensive fire control (ie.2 or 3; Table 5) So, the 0-3 ordinal scale for fire control was reduced down to a Boolean scale: 0-1 were grouped as no substantive attempt to control, 2-3 grouped as a substantive attempt to control. The result is, in effect, a variable reflecting a meaningful attempt to control a given fire, which was used to calculate the ratio of escaped fires with control measures to those without. The rate of escaped fire for each fire use type and fire control present/absent was calculated as:


$$escape_{rate_i}| \, control_i = \rho_i * \left. \left( \sum fire_{escape_i} \mid control_i \right) \middle/ \left( \sum fire_{escape_i} \right) \right. \tag{6}$$

$$escape_{rate_i}| \, !control_i = \rho_i * \left. \left( \sum fire_{escape_i} \mid !control_i \right) \middle/ \left( \sum fire_{escape_i} \right) \right. \tag{7}$$

where $\rho$ is the global mean rate of escape for each fire type, $fire_{escape_i}$ is the number of DAFI records for fire use $i$ which

describe escaped fire, and $control_i$ is a Bernoulli random variable representing the probability of the presence or absence of fire control measures.

The next step was to develop a distribution model of the *control* variable (eq. 6&7). This was done with simple classification trees. Regimes of fire governance, management philosophy and their associated degree of fire control measures emerge through

complex interactions of land users with their local socio-ecological circumstances, and policy at multiple spatial scales (Gil-Romera et al., 2011; Seijo et al., 2015; Mistry et al., 2016). Therefore, rather than parameterising for individual AFTs, the modelled distributions of anthropogenic fire regimes (AFRs) in WHAM! were used as predictor variables, representing the complex landscape-level meta-effects of interactions of multiple actors. Pyrome management was overwhelmingly used with control measures (548/565 cases in DAFI), and so data on uncontrolled pyrome management fire were too sparse to detect the

impact of control measures on escape rate. Therefore, as a simplifying assumption, all pyrome management fires were assumed to be controlled.





**Table 5: Parameterisation of escaped fire from Eq. 6&7. Baseline rates of fire escape were calculated from data in DAFI (Millington et al., 2022). The spatiotemporal distribution of the presence of fire control measures was modelled using simple classification tree models (Supplement B). Pyrome management was assumed to be controlled in all cases. Aside from pyrome management, managed pasture fires are larger (mean = 33.9ha) than other uses (mean <= 9.2ha); hence, pasture fires' more frequent escape rate is offset by a lower density per unit area burned.**

| Fire use | Fire Controlled | Baseline escape rate (%) | Impact of control | Adjusted escape rate (%) |
|---|---|---|---|---|
| Crop field preparation | FALSE | 0.06 | 2.87 | 0.17 |
| | TRUE | 0.06 | 0.35 | 0.02 |
| Crop residue burning | FALSE | 0.01 | 6.43 | 0.06 |
| | TRUE | 0.01 | 0.16 | 0.00 |
| Hunting and gathering | FALSE | 1.10 | 1.04 | 1.15 |
| | TRUE | 1.10 | 0.96 | 1.05 |
| Pasture management | FALSE | 5.10 | 1.61 | 8.19 |
| | TRUE | 5.10 | 0.62 | 3.17 |
| Pyrome management | FALSE | 0.06 | NA | 0.06 |
| | TRUE | 0.06 | NA | 0.06 |
| Vegetation clearance | FALSE | 0.95 | 3.42 | 3.25 |
| | TRUE | 0.95 | 0.29 | 0.28 |

## 2.3.2 Arson

Arson was defined as fire used deliberately to harm persons or damage property. Fires caused through carelessness such as untended campfires or cigarettes dropped from car windows were categorised as background or accidental fires (Section 2.3.3). As arson fires are lit to cause damage, they are typically not managed and cannot be considered to have an intended burned area in the same way as a pasture or crop residue fire. Therefore, rather than using burned fraction as the dependent variable in the burned area calculation, fires $km^{-2}$ $year^{-1}$ was used.



Furthermore, similar to escaped fire, arson is frequently associated with landscape-level effects, particularly conflict between land users over tenure (e.g. Suyanto, 2007). Therefore, the modelled distribution of AFRs, rather than secondary data were used as predictor variables. The impact of very inaccessible terrain such as deserts, the arctic tundras and rainforests with associated very low populations was not fully accounted for in initial model outputs. This effect was calculated, in addition to constraints described in Section 2.2.1, as:

$$Arson_{adjusted} = \widehat{Arson} * (1 - Unoccupied) \tag{8}$$

where $\widehat{Arson}$ is the output of the arson distribution model, $Unoccupied$ is the fraction of the cell unoccupied by humans and $Arson_{adjusted}$ the final calculation of the number of arson fires – adjusted for areas without human occupation.

### 2.3.3 Background fires

Background fires comprise accidental or incidental fires not captured in managed, escaped or arson fires. These include unintentional fires caused by, for example, sparks from cigarettes, forestry machinery, and from faulty powerlines and other anthropogenic infrastructure (Brennan and Keeley, 2017; Sizov et al., 2021; Bandara et al., 2023; Jenkins et al., 2023). It also includes mis-managed domestic fires and escaped waste disposal fires in urban areas (e.g. Langer et al., 2017), as WHAM! does not explicitly parameterise the behaviour of urban residents (who are assumed to occupy land but not manage it in a way that influences landscape fire). Fire density data (fires km$^{-2}$ yr$^{-1}$) were selected from DAFI where the recorded fire purpose was accidental or unknown. Using these data as a dependent variable, a simple regression tree was then developed.

### 2.4 Fire suppression

Fire suppression here refers to the extinguishing of active fires. Similar to fire control measures, fire suppression emerges from interactions local land users with policy at multiple spatial scales (Fernandes et al., 2016; Steen-Adams et al., 2017; Bilbao et al., 2019; Eloy et al., 2019) . As such, the degree of suppression was also treated as a meta-effect, calculated as a function of distribution of the four Anthropogenic Fire Regimes (AFRs) rather than individual AFTs. The AFR distribution calculated by WHAM! became the independent variables in an ordinary least squares regression, the dependent variable of which was the fire suppression indicator in DAFI. Fire suppression was recorded in DAFI on a 0-3 ordinal scale: 0 = None, 1 = Limited, 2 = Moderate or Traditional, and 3 = Intensive. To convert this to a numeric value, these ordinal values were given values (0-1) reflecting the proportion of fires extinguished (Table 6); ultimately these values and associated uncertainties will be defined during calibration of the coupled model (Figure 1).





**Table 6: Overview of DAFI data used in fire suppression model; unmanaged fires extinguished became the dependent variable in a linear model of the distribution of anthropogenic fire regimes. These initial values will be updated during calibration of the planned coupled model with JULES-INFERNO.**

| Ordinal fire suppression intensity (0-3) | Count (DAFI records) | Numeric fire suppression intensity (0-1) |
|---|---|---|
| 0 (None) | 150 | 0 |
| 1 (Limited) | 327 | 0.05 |
| 2 (Moderate or community-led) | 218 | 0.25 |
| 3 Intensive | 289 | 0.9 |

## 2.5 Model setup for historical runs

To assess and understand the model's outputs and behaviour, WHAM! was run for a historical period from 1990-2014. The rationale of this timeframe was driven by data availability. DAFI focused on 1990-2020, whilst 2014 represented the end of the CMIP6 historical run period. As noted in section 2.1, parameters of sub-models for LFS and AFT competition for land have numerical distributions derived from bootstrapping. Therefore, in model runs, 100 samples were drawn from these distributions and the mean value taken. As these distributions form a core part of the model ontology (capturing the transition zones between the niches of different land use systems), they are not a full representation of model uncertainty per se. As such, results presented below focus on the mean values of outputs of these 100 runs.

A baseline run used model inputs that took their historical values. To understand the rationale behind WHAM! outputs for managed fire, two counterfactual experiments were run and compared with the baseline run:

- LC90 (land cover 90) - in which land cover was held constant at 1990 levels; and
- LU90 (land use 90) – in which socio-economic forcing data (GDP, HDI, market access & population) were held constant at 1990 values.

As primarily an empirical model, WHAM! has only 6 free parameters; a model sensitivity analysis was conducted to fit these parameters, as described in Supplement D.



### 2.5.1 Model evaluation

Calculating burned area from unmanaged fires projected by WHAM! requires coupling with a biophysical model. Prior to that coupling (e.g. with JULES-INFERNO, as planned), evaluation of model outputs here focuses on managed fire only. Crop residue burning outputs are the easiest to evaluate, as these can be compared directly with the GFED5 crop fires product (Hall et al., 2023). Similar to the first FireMIP (which used the GFED4 & 4.1s products), this comparison was done using data for the overlapping period of WHAM! historical runs and the MODIS-era of GFED5 (2001-2014; Rabin et al., 2017). As in Teckentrup et al, (2018) pearson's correlation coefficient between WHAM! outputs and GFED5 was calculated using a square-root transformation to account for the skewed distribution of burned area. To account for differences in underlying cropland distributions that are inputs to GFED5 (the MODIS-derived MCD12Q1; Hall et al., 2023) and WHAM! (LUH2; Hurtt et al., 2020), correlations were also calculated for the proportion of cropland burned per pixel.

In addition, a broad assessment of plausibility of WHAM! managed fire outputs was made. The great majority of managed anthropogenic fires are small, and smaller than the 21ha threshold at which MODIS can reliably detect burned area (Andela et al., 2019; Millington et al., 2022). However, in GFED5, MODIS is cross-referenced against fine-scale remote sensing data from Landsat ($30m^2$) and Sentinel-2 ($20m^2$), and hence small fires are beginning to be incorporated in global-scale Earth observation. As such, between 2001-2014, GFED5 has mean burned area 457.7 Mha greater than GFED4 (800.3 Mha vs 342.6 Mha respectively, Giglio et al., 2013; Chen et al., 2023). Of course, not all of this difference is necessarily due to anthropogenic fires (van Wees et al., 2022), but comparison of the GFED4 to GFED5 increase and WHAM! managed fire outputs offers a high-level assessment of their plausibility.

Furthermore, evaluation of the sub-models for managed and unmanaged fire and fire suppression was conducted in three ways. Firstly, within-sample performance of managed and unmanaged fire parameterisations is assessed using $r^2$ for regression and AUC (area under the received operated curve) for classification. Secondly, these parameterisations are compared against out-of-sample (unseen) DAFI data. These are available due to the fragmented nature of data on human-fire interactions. For example, if the dependent variable of a fire use parameterisation was the percentage of a landcover burned, then data from papers reporting fire return interval (but not % burned area) could be used as unseen evaluation data. Thirdly, the temporal trend in WHAM! outputs was evaluated by comparison with the qualitative evaluation of temporal trend in the livelihood fire database (LIFE; Smith et al., 2022). LIFE provides assessments of whether 'subsistence-oriented' and 'market-oriented' fire use were declining, stable or increasing in the human-fire use literature. The details and outcomes of these evaluation steps using DAFI and LIFE case study data are reported in Supplement C.




## 3. Results

### 3.1 Overall model outputs

### 3.1.1 Managed fire

Over the study period of 1990-2014, modelled burned area from managed anthropogenic fires decreases from 431.9 to 419.1 Mha. In percentage terms, this equates to a 3% decline. There is substantial heterogeneity in the trend amongst fire use types. The overall modelled decline in burned area is primarily due to a decrease in fire for pasture management, which declines

20.1% from 192.04 Mha to 153.7 Mha over 1990-2014 (Figure 3). This is complemented by declines in shifting cultivation (crop field preparation) fire (31.5 Mha to 26.9 Mha) and hunter gatherer fire (23.2 Mha to 19.4 Mha). By contrast, crop residue burning increases by 17.0% from 112.0 Mha to 131.1 Mha and pyrome management fire use increases by 15.7% from 69.5 Mha to 80.4 Mha. In absolute terms, vegetation clearance fires burn the smallest area (3.4-9.1 Mha), but in relative terms, their increase is much the largest (217%), highlighting this growing environmental challenge.

Burned area from WHAM! managed fire has a coherent relationship with the difference between GFED5 and GFED4 outputs. From 2001-2014 (the overlapping period of WHAM! and the MODIS-era of GFED), the mean difference between GFED5 and GFED4 is 459.4 Mha, compared to burned area in WHAM! of 428.9 Mha (Figure 3b). Pearson's correlation coefficient between WHAM! and GFED4-GFED5 difference (r = 0.70) is greater than the mean of the model ensemble in the first FireMIP

(r = 0.65; Teckentrup et al., 2019) indicative of good performance for a first-in-class model. As noted above, comparison of WHAM! to the GFED5 to GFED4 difference is not a direct comparison of managed fire but is provided as an initial assessment of WHAM! output plausibility. Direct evaluation of crop residue fires is provided in Section 3.2 below.

Beneath the global trends in managed fire, there is also substantial spatial heterogeneity (Figure 4). At the continental scale,

the decline in pasture management fire dominates in South America, declining from 55.31 Mha in 1990 to 25.15 Mha in 2014, leading to a decline in overall managed fire from 102.14 Mha to 71.12 Mha (Figure 5). By contrast, in Africa pasture fire *increases* by 6.51 Mha, whilst in Asia a decrease in pasture fire of 9.83 Mha is more than offset by a steep increase in crop residue burning of 18.17 Mha.





**Figure 3: Temporal trends in managed fire. A) burned area by fire use type, and B) total WHAM! managed fire compared against the difference in burned area in GFED5 & GFED4. Shading in A) shows the 5th and 95th percentiles of the outputs from WHAM! parameters' numerical distributions. Overall, pasture fire accounts for both the largest amount of fire and the largest absolute decline. In cropland systems, shifting cultivation fire and residue burning exhibit opposite trends. Whilst vegetation clearance fire is small in absolute terms, it shows the largest relative increase over the model period. Key: CFP = Crop field preparation, CRB = Crop residue burning, HG = Hunter gatherer, Pasture = Pasture management, Pyrome = Pyrome management, VC = Vegetation clearance.**





**Figure 4: Global model outputs for managed fire in 1990 & 2014 grouped by land cover. Forestry and non-extractive fire use types are grouped together as this will be how they are interpreted by JULES-INFERNO. Maps highlight the decline in pasture fires in South America. Conversely, pasture fire increases in Sub-Saharan Africa. Crop fires increase in Northern India, South Asia and modestly in South America, but decline elsewhere. Vegetation fires cover those for hunting and gathering, pyrome management and vegetation clearance.**





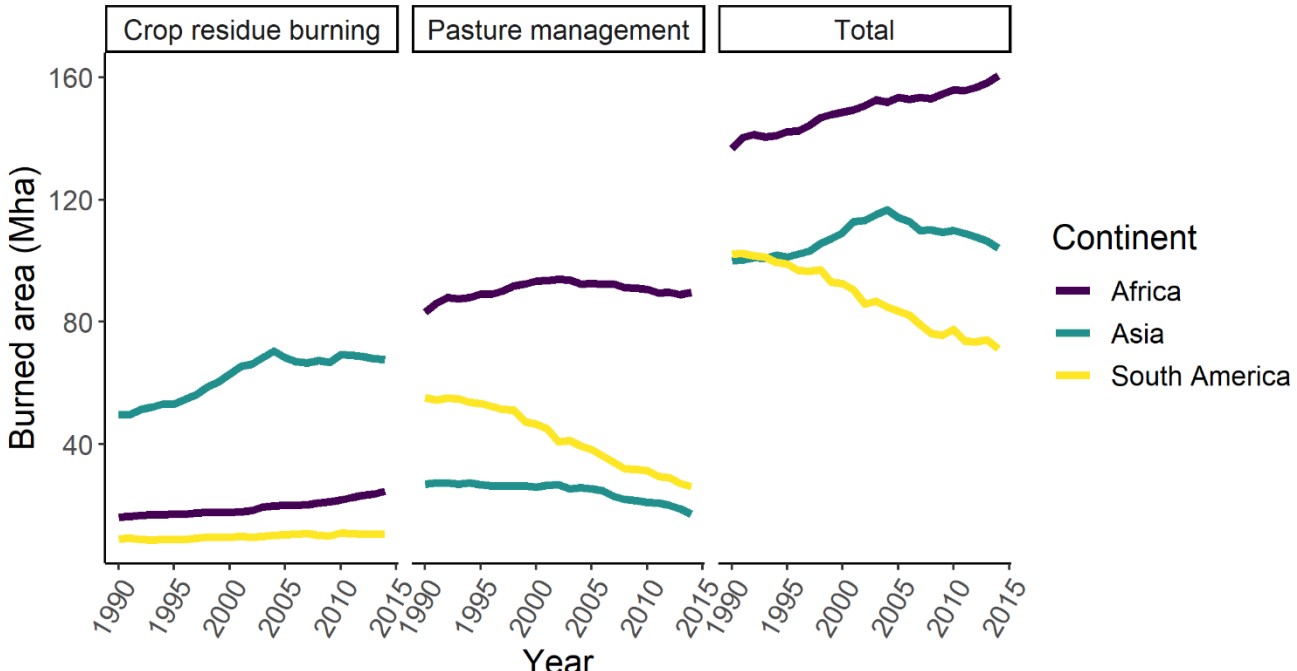

**Figure 5:** **Managed fire burned area for the two dominant modes of managed fire & total managed fire for the three continents with largest burned area from managed fire. Whilst the global declining trend in pasture management fire is dominant in South America, in Africa pasture and crop residue fires contribute to an overall slight increase. Similarly, in Asia a decline in pasture fire is offset by a marked increase in crop residue fires.**

### 3.1.2 Unmanaged fire

Whilst burned area from managed fire modestly decreases by 3% globally, the picture from unmanaged fire is mixed. Arson and accidental anthropogenic fires both increase (Figure 6): the background rate of accidental fires increases 24.8% whilst the rate of arson increases 17.7%. By contrast, the number of escaped fires decreases (-8.6%), mirroring the decrease in burned area from managed fire. However, until WHAM! is coupled with a DGVM, it will not be possible to deduce if this has led to increase in burned area from unmanaged anthropogenic fire. This consideration is particularly important given the distribution of unmanaged fires is seemingly clustered around wildland urban interface areas (Figure 6b), meaning that many of these ignitions will likely be extinguished through industrialised fire fighting (Millington et al., 2022).





**Figure 6: Unmanaged fire outputs as fires yr⁻¹ km⁻²: A) temporal change and B) spatial distribution in 2014. The rate of unmanaged fires increases over the modelled period. However, this increase is clustered towards Wildland Urban Interface areas (visible as spatial anomalies in B), and the impact of this on burned area will only be clear after coupling with JULES-INFERNO.**




### 3.1.3 Fire suppression

Overall, modelled fire suppression increases from 1990-2014, particularly in South America, South and South-East Asia (Figure 7). The relationship of suppression intensity to numbers of unmanaged fires will be determined through the planned coupling with JULES-INFENRO.


**Figure 7: Fire suppression intensity in 1990 & 2014.**





## 3.2 Evaluation with GFED5 crop data

WHAM! outputs for crop residue burning are in broad agreement with GFED5 crop fires (Figure 8). Correlation (Pearson's r) is 0.673 for burned area per pixel, and 0.665 for rate of cropland burned per pixel. WHAM! crop residue outputs project more burning than GFED5, with an annual mean of 129.2Mha over the overlapping period (2001-2014) compared to 87.6Mha for GFED5. The main continent driving disagreement is Asia: 67.8Mha in WHAM! compared to 31.2Mha in GFED5 (Figure 9).

WHAM! and GFED5 disagree on the trend of global crop fires, with WHAM! projecting a global increase and GFED5 suggesting a decrease (Figure 9). At the continental-scale, WHAM! and GFED5 agree on the trends in Europe and North America (decreasing). However, WHAM! projects gains in Asia (GFED decreasing), as well as increases in Africa (GFED decreasing). By contrast WHAM! exhibits contrasting trends between crop residue fires and other managed fires. For example, in South America and Asia, WHAM! residue fires and other managed fires are negatively correlated (r = -0.91, -0.74

respectively), whilst GFED5 outputs for crop fires and the overall regime are positively correlated in all cases.

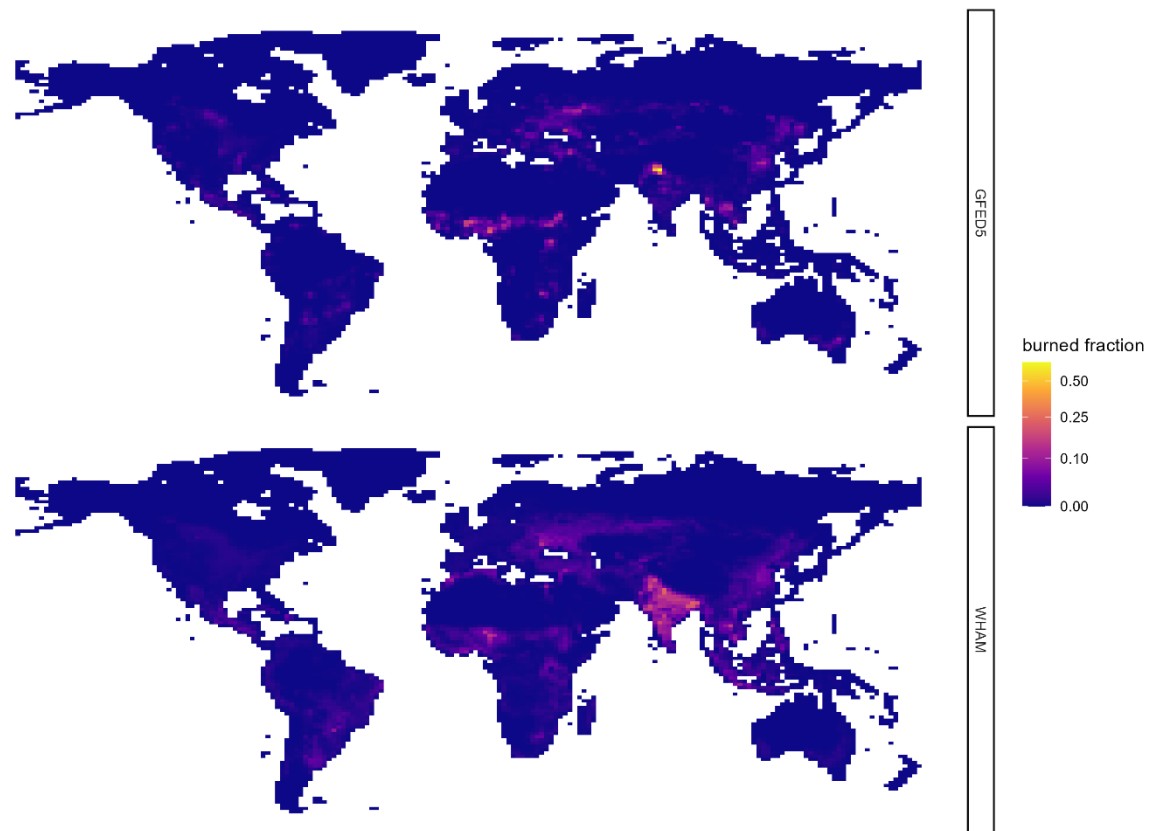

**Figure 8: Comparison of WHAM! crop residue burning outputs and GFED5 crop fire outputs in 2014. Whilst overall coherence is good, disagreements are most evident in Northern India.**





**Figure 9: Continent-scale trends in burned area for WHAM! crop residue fires and GFED5 crop fires. The biggest area of disagreement in burned area is in Asia. Conversely, whilst in Africa burned area is similar, WHAM! & GFED5 show opposite temporal trends. Note the differing y-axis values.**





### 3.3 Understanding model behaviour

Counterfactual experiments reveal divergent impacts between land cover change and changes in land use intensity. In the LC90 experiment, where land cover was held constant at 1990 levels, managed fire declines more starkly than in the baseline model run (431.94 to 388.74 Mha; Figure 10). By contrast, LU90 (land use intensity - and therefore AFR - constant at 1990 levels) leads to an *increase* in overall managed fire from 431.94 to 472.10 Mha.

The effects of land cover and land use intensity on human fire use have clear spatial patterns (Figure 10b). In LU90, the increase in fire over the baseline scenario is most evident in South America, highlighting the importance of land use intensification in this continent as a driver of changing fire regimes. Similar increases over the baseline are present in North-eastern China and Mexico. By contrast, in Northern India, the LU90 counterfactual leads to decreased fire against the baseline, indicating land use intensification has led to increased fire use. This finding fits previous analyses of crop residue burning in
the Indo-Gangetic Plain (Liu et al., 2019; Sembhi et al., 2020). The LC90 (constant land cover) counterfactual has more consistent global effects, with decreases in fire over the baseline observed in regions with large amounts of extensive livestock farming - particularly Madagascar, the Guinean Savana, and Southern Brazil.

    Divergent trends between land use and land cover change on human fire use point to similarly divergent socio-economic
drivers across differing modes of fire use (Figure 11). For example, at global scale, population density seems to be associated with increased crop residue burning (r = 0.31). By contrast, population density has a more ambiguous effect on pasture management fires (r = -0.05), the distribution of which is negatively correlated with socio-economic development (as measured by the HDI; r = -0.47).

Similarly, across the three continents with the highest rates of agricultural fire – Africa, Asia, and South America – increased HDI consistently leads to decreased fire use for pasture management (Figure 11b). However, in these three continents, increased HDI can lead to either increased *or* decreased fire use for crop residue burning: at a mean HDI of ~0.6 such fire use increases substantially in Asia but decreases in South America. Possible process-based explanations of this trend are offered in the discussion (Section 4.1).






**Figure 10: Global burned area from managed fire under counterfactual scenarios. A) global trends 1990-2014; B) change in burned area between counterfactual and baseline scenario in 2014. Key: LC90 – land cover held constant at 1990 levels; LU90 – land use intensity held constant at 1990 levels.**



**Figure 11: Drivers of managed burned area for the two modes of anthropogenic fire use with largest global burned area: A) by pixel, and B) by continental mean. Population density marginally increases the rate of crop residue burning and has an unclear impact on pasture management. Similarly, the human development index (HDI) has a similarly complex relationship with fire use: increased HDI consistently leads to decreased pasture fire, but can lead to divergent outcomes for residue burning.**



## 4. Discussion

We have presented WHAM!, the first global behavioural land system model of human fire. The ultimate intention is to couple WHAM! with the JULES-INFERNO DGVM. Here, WHAM! has been presented in standalone form. Therefore, the discussion focuses on managed fire, which can be independently evaluated without input from JULES-INFERNO.

### 4.1 WHAM! outputs

WHAM! outputs suggest burned area from managed anthropogenic fire declined by 12.8Mha from 1990 to 2014. This is driven
by decreased fire use for pasture management particularly in South America (Figure 5), and complemented by declines in crop field preparation (shifting cultivation) and hunting and gathering fire. By contrast, fires for crop residue disposal increased by 19.1Mha. These divergent trends broadly align with Smith et al., (2022), whose global meta-analysis found that 'subsistence-oriented' fire uses are declining, whilst 'market-oriented' fire uses are increasing (Supplement C). Globally, empirical data shows cropland fire uses produce the smallest anthropogenic fires (mean = 3.9ha; Millington et al., 2022). Therefore, outputs
from WHAM! are also consistent with initial results from the Global Fire Emissions Database version 5 (GFED5), which suggest smaller fires – which are principally anthropogenic – have declined less than larger ones (Randerson et al., 2022). Taken together, these recent advances spanning literature meta-analysis, remote sensing - and now modelling - suggest changes in anthropogenic fire use are contributing to the observed global decline in burned area (Andela et al., 2017).

Counterfactual experiments and analysis of the drivers of pasture management fire in WHAM! demonstrate that the modelled decline in pasture fire is primarily due to land use intensification (Figure 10). This finding matches real-world observations. For example, the rapid pace of land use intensification in South America was documented by Silva et al., (2017), who attribute changes to the 'telecoupled' system of soybean production in response to increased demand for meat. Furthermore, this process of declining fire under increased land use intensity was explored in the field experiments of Cammelli et al., (2020), who find
that increased capital investment discourages fire use as a management strategy because fire increasingly becomes a risk to machinery, irrigation, and other capital investments.



By contrast, the relationship between increasing societal development (represented by the Human Development Index) and crop residue burning is more ambiguous (Figure 11). In Africa (HDI: 0.3-0.5), increased HDI seems to increase crop residue
burning, consistent with land use intensification driving this practice. However, at intermediate (0.6-0.85) levels of HDI increased development can have divergent impacts on residue burning, notably between Asia and South America. It is possible that farm size, and therefore the production system, plays a role here: large soybean farmers in South America engaged in formal, legalised supply chains are somewhat likely to comply with anti-burning as well as general fire management legislation (Soares-Filho et al., 2014; Villoria et al., 2022). By contrast, in Asia, and the Indo-Gangetic Plain in particular, high rural
population density and small average farm size entails that production is dominated by small-holder farms with associated informal supply chains (Birthal et al., 2017), making environmental enforcement more challenging (Bhuvaneshwari et al., 2019; Liverpool-Tasie et al., 2020). In WHAM!, this difference is seemingly captured through the impact of population density, which features in the classification tree for the small-holder land fire system (Perkins et al., 2022).

WHAM! suggests human fire use can either increase or decrease with increasing population, in ways that are highly heterogenous and specific to the rationale of the underlying land system and associated modes of fire use (Figure 11). At global scale, crop residue burning slightly increases with population density, but there is a weak negative relationship with pasture fire. Taken together, these complexities illustrate the shortcomings of relying on a single function of population density to capture the full spectrum of human-fire interactions globally (Teckentrup et al., 2019) and the benefit of taking a categorical
approach to developing functions for representing human activity.

Therefore, whilst WHAM! is, at root, a relatively simple empirical model, it captures complex dynamics amongst the drivers and spatiotemporal distribution of human-fire interactions. This is further highlighted by the use of anthropogenic fire regimes (AFRs) to capture emergent or landscape-level effects. For example, arson fires become 17.7% more frequent over the period
1990-2014, driven by WHAM!'s representation of land tenure conflict through the transitional AFR. By contrast, the number of escaped managed fires declines (-8.6%). Using the AFRs as predictor variables in parameterisations of arson and fire control behaviours (which have an important influence on escaped fire) results in high predictive accuracy (AUC >= 0.8; Supplement C). As such, we can conclude that use of the AFRs allows differing trends in sources of unmanaged fires to be identified (Figure 7), further highlighting the shortcoming of relating to anthropogenic fire directly to population density without
considering the global diversity of human-fire interactions.





## 4.2 WHAM! evaluation and limitations

Comparison with GFED5 cropland fires suggests WHAM! credibly reproduces global patterns of crop burning. However, there are some disagreements between WHAM! and GFED5 crop fires, notably in terms of the spatial distribution of such fires
in India, and the temporal trend in sub-Saharan Africa. Evaluating the distribution of the land-fire systems in WHAM!, Perkins et al., (2022) find that the crop fire hotspots in Northern India and in North Eastern China were also spots of disagreement between WHAM!'s distribution of land fire systems and the human appropriation of net primary production – an independent measure of land use intensity (Haberl et al., 2007). As such, this difference may point to underlying difficulties in representing land use transitions through a single transitional fire regime type per land system, when in reality multiple trajectories of
intensification (and de-intensification) can occur, even within apparently similar land system types (Lambin and Meyfroidt, 2010; van der Sluis et al., 2016).

However, it is important to note that the GFED5 cropland product is the first of its kind - and *is itself a model* relying on empirical scaling factors to infer burned area per active fire detection in cropland areas (Hall et al., 2023). These scaling factors
for rice burning were developed from fieldwork in Ukraine (Hall et al., 2021), where the mean field size of 40ha is much larger than the smallholder fields in Northern India (~1ha). Hence, a single active fire detection in Northern India could very possibly equate to more real world burned area than in Ukraine, indeed as is suggested by the work of Deshpande et al., (2022). Increased spatial coverage of ground-truthed, landscape-level remote sensing work of cropland burning is needed to advance understanding further.


As with any empirical model, WHAM! is inherently limited by the strengths and weaknesses of its underlying data. There were three central ways in which such uncertainties in the Database of Anthropogenic Fire Impacts (DAFI, against which WHAM! was parameterised) came through in model outputs, which are discussed in turn below. The first of these was in the parameterisation of fire use in nomadic land use systems, particularly shifting cultivation and pastoralism. The mediocre
performance of the underlying models is indicative of the difficulty of quantifying fire regimes produced by such land use systems (AUC = 0.623, $r^2$ = 0.07). For example, shifting cultivation is challenging to study with remote sensing, not only as it is semi-nomadic, but also due to the complex spectral signals produced (Jiang et al., 2022), which make differentiating between fields and early-successional regrowth a substantial challenge (Heinimann et al., 2017). As a result, the fallow period was typically used from field studies as a proxy for fire return period. This involved assumptions about the duration of cultivation
after fallow; here assumed to be two years – yet this can vary from 1-5 years (e.g. Maharani et al., 2019). Pastoralism is also challenging to study with remote sensing, due to the difficulty of tracking pastoralists' location across the large areas over which they may migrate seasonally (Nelson et al., 2020). However, it should be noted that these fire uses represent a small amount of global burned area: burned area from shifting cultivation was 26.9Mha, whilst migratory pastoralist fire accounted for just 18.4Mha of burned area in 2014.





Secondly, more structural sampling biases within DAFI led to the need for top-down constraints being applied to the bottom-up parameterisation of fire uses. These arose firstly as DAFI did not sample very arid environments (the vegetation constraint); and secondly, because DAFI under-sampled more developed contexts (Perkins et al., 2022). However, sensitivity analysis demonstrated that WHAM! is not overly sensitive to the resulting free parameters – with burned area outputs varying by a maximum of ±4.4% (Supplement D). Furthermore, the most sensitive parameter was not for a top-down constraint, but the

'theta' parameter, which sets the threshold at which a given AFR's competitiveness score was set to 0 (Supplement D). This is somewhat analogous to the 'giving-in' parameter in the CRAFTY land system model, which determines when a land use type becomes uncompetitive (Murray-Rust et al., 2014). CRAFTY is highly sensitive to this parameter (Seo et al., 2018), which is an uncertain function of agent behaviour. This seems a strength of the empirical approach taken here, as it appears less reliant on uncertain abstraction.


A third fundamental issue arises from the coarse spatial resolution of WHAM! and how that relates to the data available for parameterisation. Specifically, due to the intended coupling with JULES-INFERNO, the spatial resolution of WHAM! and DAFI case study data are substantially different: the median WHAM! cell is seven times larger than the median DAFI case study (24,684 vs 3,508 km$^2$). However, this is likely a large underestimate of the true discrepancy. Only 30% of case studies

in DAFI reporting pre-industrial AFRs quantified their study area, compared with 82% of industrial AFR case studies (Perkins and Millington 2021). The mean reported study area for industrial AFR case studies is 53 times larger than for the pre-industrial AFR. This trend is likely even more acute for LIFE (Smith et al. 2022), as it focuses on 'livelihood fire' – which broadly corresponds to the pre-industrial and early transitional AFRs in WHAM!. However, case study area is not recorded in LIFE (Smith et al., 2022). The consequence of this disparity in spatial resolution is seen in the evaluation of model outputs against

unseen case study data in DAFI and LIFE: WHAM! captures macro-scale trends, but in its first iteration does not fully capture more detailed trends at the case-study level (Supplement C).



## 5. Conclusion

This paper has presented WHAM!, the first global behavioural land system model of present-day human-fire interactions.
WHAM! is designed to be coupled with dynamic global vegetation models – JULES-INFERNO in the first instance. As such, full evaluation of WHAM! will only be possible once this coupling is complete. However, here we demonstrate that independently, WHAM! is effective at capturing both spatial and temporal trends in Earth observation data of burned area. Evaluation of WHAM! with GFED5 crop fires reveals strong spatial coherence, whilst WHAM! managed fire outputs project a decline in burned area between 1990-2014, driven by land use intensification and declining fire use for extensive livestock
farming. Burned area from managed fire outputs in WHAM! is close to the difference in global Earth observation data with (GFED5) and without (GFED4) small fires, establishing the broad plausibility of model outputs.

Drivers of human fire use in WHAM! are divergent across differing fire use types and spatially heterogenous, pointing to the fundamental limitation of globally-uniform population-based approaches, which are widely used in existing global fire models.
Additionally, use of landscape-level anthropogenic fire regimes proves an effective way to capture influences that emerge from interactions between land users: this includes fire suppression, the extent of control applied to managed fires, and arson. Overall, the diversity of human-fire interactions and their divergent spatiotemporal trends highlight a fundamental need for consideration of the socio-ecological drivers of fire regimes in dynamic global vegetation models.

**Code and data availability**

All code and data to run WHAM! version 1.0 are made freely available online via Zenodo (Perkins et al., 2023a, 2023b). This includes an installation and user guide. For convenience, Github installation is also available at: https://github.com/OliPerkins1987/Wildfire_Human_Agency_Model.

**Author contribution**

All authors contributed to model conceptualisation and methodological design from their contrasting disciplinary perspectives; OP wrote the software, whilst MK ensured model structure supported integration with JULES-INFERNO; OP & CS were responsible for data curation. AV secured funding. JDAM provided project oversight and leadership. OP wrote the original draft, whilst all authors contributed to editing and revisions.

**Competing interests**

The authors declare that they have no conflict of interest.

**Acknowledgements**

The authors wish to thank Davide Lomeo and Toby Wainwright for their kind help with cross-platform compatibility testing. The authors also wish to thank Calum Brown, Tamsin Edwards and Colin Prentice for their helpful comments and suggestions.



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
