# Peer review of "A global behavioural model of human fire use and management: WHAM! v1.0"

_EGUsphere, 2023_

## Author Comment (AC1)

**Reviewer response: *A global behavioural model of anthropogenic fire use and management: WHAM! v1.0**

We wish to thank both reviewers of our manuscript for their thoughtful comments and recommendations. We respond to Carolina Ojeda Leal as reviewer A and Sam Rabin as reviewer B. Please note that in the revised version of the manuscript, we have reordered the results section, as suggested by Reviewer B. We have noted present and previous figure numbers where relevant.

**Reviewer A**

Comment: First of all, the acronym WHAM is an excellent catchphrase, congrats on that. This paper presents an updated version of DGVM to understand human-fire dynamics globally, which is a valuable source of future fire trends. I suggest adding the most recent paper of Fischer et al. (https://doi.org/10.1016/j.crm.2023.100573) and the paper of Smith et al. (https://doi.org/10.1093/biosci/biv182) to discuss the difference between the terms proposed by those authors: Global Adaptation Mapping Initiative (GAMI) fire-human adaptation database and firescapes.

RESPONSE: Thank you for these comments and reflections. We share your enthusiasm for our model's name.

In many ways, WHAM! represents an early attempt to deliver modelling capacity in support of the research agendas set out in both the Fischer and Smith papers. Rather than refer to the Smith paper, we chose to frame our work in the context of the more recent review of Schuman et al., (2022). The Schuman paper addresses similar questions of the lack interdisciplinary in fire science as Smith, but also specifically mentions representing managed fire in global-scale models as a key research challenge.

Similarly, Fischer describe how combinations of environmental and socio-economic factors drive the implementation of climate adaptation measures (Fischer Figure 1). Developing model capacity to explore and evaluate climate change adaptation measures is a major long-term driver of the creation of models such as WHAM! (see Arneth et al., 2014; Perkins et al., 2023). For context, this is a part of the development of a relatively young field of science known as 'Human-Earth system modelling'. However, because this journal is primarily focused on development of the biophysical side of vegetation and fire models, we focus on WHAM!'s contribution within this research agenda, rather than this broader (really important!) transdisciplinary context.

Furthermore, I wonder what is the difference between the land-fire systems (LFS) proposed here to classify the anthropogenic fire regimes (AFRs) with the categories proposed by Blondel in his paper (https://doi.org/10.1007/s10745-006-9030-4) or traditional land use land cover obtained with GIS used in other articles (https://doi.org/10.5194/nhess-21-3663-2021).

RESPONSE: Our AFRs draw on the work of Stephen Pyne, and WHAM! is in many ways an implementation of his theoretical framing of human-fire interactions in a simulation model. The first study that you point to is in strong alignment with Pyne's "Second-fire" in which widespread use of fire as a land management tool has (re-) shaped the structure of vegetation and landscapes globally. The processes analysed in the second study you point to are captured in our model through representation of abandoned forestry plantations (Table 1) as well as an attempt to represent increased fire risk in the suburban regions of cities (Figure 9).

We describe how our AFRs reflect Pyne's 1st, 2nd and 3rd fire in some depth in both Millington et al., 2022 and Perkins et al., 2022. As such, we here focus on how WHAM! operationalises this framework. However, we now have made this relationship of our AFRs to Pyne's work explicit in this manuscript at line 136.

Comment:  Also, please add in line 525 another explanation of land use intensification in South America, it is not just clearing the land for cattle. For example, in Chile, we have increased over the decades the available land for pyrophitic species (like pine and eucalyptus) and urban sprawl.

RESPONSE: A useful point, thank you. Our point here is not that the expansion of land use area (extensification) is changing fire regimes, as would be suggested by urban sprawl or land clearing for cattle. Rather, we are arguing that *intensification of existing agricultural lands* – for example the conversion of extensively farmed pastures into intensively farmed soybean systems – is leading to a decrease in fire use. The point you make about forestry plantations is a good one – we do indeed have a representation of forestry plantations, and indeed abandoned forestry plantations in the model (see new supplementary information E, which contains a complete set of maps of these).

As we note in the main text, our model can capture certain aspects of fire regimes as a standalone model, but requires integration with biophysical vegetation models (i.e. DGVMs) in order to provide a complete picture of the drivers of fire. Forestry plantations of pyrophytic species, and indeed abandoned forestry plantations, might be expected to play an important role in driving *unmanaged fires* (i.e. wildfires). However, managers of such plantations typically do not make widespread proactive use of fire – instead adopting an exclusionary mindset. We will not be able to assess the role of pyrophytic tree species until coupling of our model with a DGVM has been completed and presented.

Lastly, the paper was written in a highly technical language that was hard to read, and therefore, WHAM will be hard to replicate for scientists in other countries.

RESPONSE: This is an important point.  We are aware that our work is dependent on expertise and knowledge from a diverse range of stakeholders and fire practitioners from many countries. There is a danger of researcher extractivism, where our work is not readily usable by those whose knowledge ultimately underpins it. We have been participating in workshops with researchers and fire practitioners around the world to share the outputs of another global synthesis of knowledge around human-fire interactions, and we will look at doing similarly for WHAM! However, here, whilst keeping technical jargon to a minimum, we think it is right that we are trying to be precise and detailed in how we communicate our method and findings. We are here trying to document our model to other scientists such that they could, in theory, reproduce it from scratch.

**Reviewer B**

I read this manuscript with great interest, as I'm pondering how to include management fires in a DGVM myself. DAFI and WHAM! represent huge leaps forward in this area. I say this because I don't want my questions and comments to come off as critical for no reason! Rather, I have so many because I'm so interested in this subject and think WHAM! is really important.

RESPONSE: Thank you for this extremely helpful, thorough and fair review.

QUESTION: - Lines 193-196: What exactly is the difference between tendency and extent? How were they separately parameterized?

RESPONSE: Thanks, this is a helpful question. We think the previous text was not clear and have revised it substantially, (now lines 201-207):

> We parameterise each AFT's decisions to use these forms of managed fire in two stages. Firstly, AFTs decide whether to use a given type of fire; this was parameterised using Boolean fire presence or absence data from DAFI. Secondly, we represent the spatial extent of an AFT's use of a given fire type, in locations where they choose to use it; this was parameterised using burned area data from DAFI. Separating a decision to use fire for a given purpose (hereafter 'fire use tendency') from the burned area generated where it is used (hereafter 'fire use extent') enables WHAM! to capture important nuances in human fire use decision-making. For example, DAFI data show that state land managers such as the US forestry service are typically fire averse and so have a low fire use tendency; but where they do use fire, for example in protected areas or other sparsely populated regions, they may burn large areas (Millington et al., 2022).

We hope this captures better what we mean by tendency (Boolean) and extent (burned fraction) and how they were parameterised differently.

QUESTION:  Line 305: What "initial model outputs" are being referred to here?

RESPONSE: Thanks, the language used here was misleading. We were just referring to the unoccupied correction being added after visual inspection of outputs the first time we pressed 'go'. On reflection, this is largely because WHAM! stores the unoccupied fraction as a land use system and not as an AFR, and as such this amounted to a coding error corrected during the development process. We have amended the text for clarity (lines 324-327).

QUESTION:   Lines 323-324: It might be worth mentioning here that fire *control* applies only to managed fires whereas fire *suppression* applies only to unmanaged fires. (This is clear from the very helpful Fig. 1, but it could use reiterating.)

RESPONSE: Thanks, we have re noted this here (now lines 344-345).

QUESTION:    Lines 359-360: This says that "evaluation of model outputs here focuses on managed fire only," but Sects. 3.1.2 and 3.1.3 look at, respectively, unmanaged fire and its suppression.

RESPONSE: Yes, we share all model outputs, but are only able to provide independent evaluation of our burned area outputs. The language here has been amended to be clear about this (now lines 385-386.

QUESTION:     Lines 382-383: Isn't fire return interval typically considered just the inverse of annual burned fraction? In which case this wouldn't be an independent evaluation.

RESPONSE: Yes; the key point here is that many papers in DAFI report burned area % or fire return interval, but not both. Hence, if we used DAFI data for burned area for developing a submodel, DAFI case studies which reported fire return period, but not burned area, could be used independently for evaluation. We have amended the text slightly for clarity.

QUESTION:    - Lines 485-487: Does this mean that "extensive livestock farming" occurs outside the LUH2 "pasture" area? If not, how does land cover affect managed fire used there?

RESPONSE: Yes, we have livestock farming on both LUH2 pastures and LUH2 rangelands. Rangelands span hugely different biophysical niches, with correspondingly very different stocking rates and availability of biomass for fire. Hence, we correct for this by representing rangelands which are sparsely occupied. This is described in Supplementary B1.3; a flag pointing to this is added at lines 217-218.

QUESTION:    Fig. 11:

   - Subplot A shows "crop" (presumably residue burning + field prep) whereas subplot B shows only crop residue burning. Why?

   - What are the individual dots in subplot B? Each continent in each year? If so, what year range? and does subplot A also include individual years?

   - Caption says that "increased HDI consistently leads to decreased pasture fire," but this isn't true for Africa—a pretty important continent when it comes to pasture burning! (Similar issue with lines 495-496.)

RESPONSE: Thanks, we have corrected the legend to Subplot A, which shows only residue burning for clarity. For Subplot B, we have added further information to the caption to clarify that each dot is the mean value for the continent from 1990-2014. This is a good point re: Africa and pasture fire. We have edited the figure caption and text (now lines 560-562) for clarity.

For Subplot A, we noted in revising this figure that the previous version had shown only values from 2014. In retrospect, we think it makes sense to show all values across the model run period and have updated the figure and caption appropriately. The correlations presented in the text between population density and HDI have been updated in a similar way. This is now Figure 12.

- Lines 481-482: What do you mean by "land use intensification"? Same for lines 525-531. LU90 doesn't seem able to distinguish between extensification (which is usually defined as increasing area of a land use type) vs. intensification (increasing management inputs/technology on existing cropland/pasture/etc.).

   - ... Actually, I think I may have misinterpreted LU90. Typically when I read "land use," it's in the context of land use *area*, but I think that's included in LC90 instead. (Typically when I read "land cover" it refers to the areas of different types of vegetation, regardless of whether e.g. a hectare of grassland is "natural" or grazed by livestock.) I think a more complete explanation of what you mean by "land use" vs. "land cover" is warranted; note that they seem to be conflated in Table 3 Column 2. I also suggest changing the name of the LU90 experiment to something like SE90 (socioeconomic conditions 1990) instead.

   - Still, the LU90 counterfactual doesn't seem able to distinguish between what I understand as "land use intensification" vs., say, increasing regulatory control of burning.

RESPONSE: Yes, this is useful clarification of terminology. From our perspective, land use can encompass intensity – including choices around whether to use machinery, chemicals or fire - as well as land cover change. However, as your rightly point out, changes here could be due to multiple effects, including policy. Hence, we have renamed this scenario 'SE90' or socio-economic 90 as you suggest. This is a more straightforward and precise reflection of the scenario set up.

QUESTION: How were the free parameters for vegetation constraint and AFR effect parameterized?

RESPONSE: Both of these parameterisations sought to correct for identified biases in the DAFI data (Perkins et al., 2022 – Figure pasted below for convenience). I.e. DAFI (and the literature generally) under-represents wealthier regions, where there is less fire use, and does not sample regions where fire is biophysically impossible. In both cases, our goal was to correct for implausible outputs that arose from these known biases, whilst still allowing for the great majority of WHAM's outputs to be based on a bottom-up and empirically-grounded approach. We took steps to address these biases in the ways we sampled DAFI data, i.e. by upsampling DAFI case studies in lower potential evapotranspiration regions and downsampling lower HDI case studies (described in Perkins et al., 2022) - but in some cases the sparse data in DAFI meant that residual bias was unavoidable. A second consideration was that beyond DAFI, there weren't any systematic data sources against which to do model or function discovery. Remote sensing data do not distinguish between managed and unmanaged fire sources. As such, we kept the functional forms as simple as possible.

The outcome of this simple approach that sought to limit the impact of free parameters on model outputs is evident in the model's relative insensitivity to these free parameters (Supplementary D), compared to the uncertainty generated by resampling DAFI (shaded areas in Figure 6). We have clarified the quite narrow role of model free parameters in the main text (lines 222-246), as well as showing the impact of these parameterisations in Supplementary D. In addition, in a new section on limitations, lines 674-684, we have outlined how the AFR constraint could be replaced in a future version of WHAM with a process-based representation of fire policy.

[Figure]

Figure R1: Biases in DAFI (from Perkins et al., 2022). Bars give the proportion of DAFI case studies appearing in each quartile of two global reference data sets (HDI and potential evapotranspiration). DAFI undersamples very low ET environments, typically where fire is not biophysically possible. DAFI also oversamples low HDI locations.

QUESTION:    Surprising trends in Africa:

   - Lines 411-412: How does the modeled increase in Africa pasture burning square with Andela et al. (2014)?

RESPONSE: Good question. A few things to keep in mind on pasture fires:

1) Over the MODIS-era (2001-2015), WHAM! pasture fire projections actually marginally decrease in Africa. Although, the difference between 1990-2015, (and indeed between 2001-2015) are very slight.
2) The median pasture fire size in DAFI (14ha) is below the level at which MODIS can detect burned area, and hence these are probably not reliably detected in Andela et al..
3) An increase in pasture fires might lead to a decrease in other types of fire due to landscape fragmentation, particularly where it contributes to patch-burning systems in which this is an underlying goal of overall fire use (e.g. Laris 2002).
4) Without pre-empting future work – in the prescribed periodic coupling of WHAM! with JULES-INFERNO, burned area in the most flammable regions of Sub-Saharan Africa is largely dominated by unmanaged fires.
5) Our version of WHAM! parameterised with Earth observation data (Appendix A), has sharper declines globally in pasture fire.

Hence the contribution of changes in pasture burning in Africa to declining burned area is far from clear. Overall, our interpretation of these results would be that changes in human fire use have likely contributed to the decline in burned area in South America but may not have done so in Africa. We think this is broadly in line with findings of Archibald et al., 2012 that in very flammable areas, the overall burned area is determined more by fuel connectivity than number of fires.

We have added a short line to the first paragraph of our discussion indicating that whilst human fire use (including pasture fires) has likely contributed to declining burned area, there are now multiple sources (including GFED5) suggesting it may have done so at a lesser rate than the overall fire regime.

   - Fig. 9: Same question for crop fires—or are they even really considered in Andela et al. (2014)?

RESPONSE: Crop fires are not reliably captured in the Andela analysis, particularly given the middling performance of GFED4.1s in capturing small fires. In GFED5, there is a new crop fires algorithm, which we use to evaluate our model. However, there is good reason to believe that it does not fully separate crop fires from wider vegetation fires, including in sub-Saharan Africa (Figure R2 below). From personal communication with the developer of the GFED5 crop fire algorithm, this is probably because of the MODIS land cover class for 'cropland mosaic', which contains other unmanaged vegetation. (See lines 629-635 in the main text).

In summary, we are confident that there are good process-based reasons to believe crop fires are very likely increasing in Africa. As chemical fertilisers and mechanised harvesting are introduced, volumes of residue produced are both in excess of what can be used practically and burdensome to gather. In hand-harvested, lower yielding systems using manuring, residues may be bailed and fed to livestock. However, the overall impact of cropland conversion in Africa is very likely negative on burned area - increases in burned area from residue fires are probably more than offset by decreases to unmanaged fires caused by landscape fragmentation from cropland conversion.

[Figure]

Figure R2: WHAM! and GFED crop fires as a proportion of total burned area (GFED5) and total managed fire (WHAM!). WHAM! is able to show diverging trends between crop fires and other managed fires, whereas the crop fire signal in GFED5 seems to track the trajectory of other fires very closely.

Comments

Comment: Lines 95-96 say that "the parameterisation of WHAM! presented in the main text takes relevant biophysical input variables from JULES model outputs," and Table 3 includes what seem to be JULES outputs from Best et al. (2011) and Clark et al. (2011) (although citations for those are missing from the reference list, so I can't be sure). But then that seems to be contradicted in some places. I think you're saying that the presented results are with WHAM! "offline" (or as you say, standalone), whereas in the future you hope to run JULES and INFERNO together at the same time. I might be getting confused because I consider what you have here to be a "one-way" coupling, whereas you hope to move to a "two-way" coupling. See Robinson et al. (2018). Modelling feedbacks between human and natural processes in the land system. Earth System Dynamics, 9(2), 895–914. doi: 10.5194/esd-9-895-2018. Anyway, here's where I got confused:

   - Lines 334-335 (Table 6 caption): "These initial values will be updated during calibration of the planned coupled model with JULES-INFERNO."

   - Lines 359-360 say that the coupling hasn't happened yet.

   - Lines 448-449: "The relationship of suppression intensity to numbers of unmanaged fires will be determined through the planned coupling with JULES-INFENRO." (Note "INFENRO" typo, and that this sentence is confusing anyway—not sure what it's trying to say.)

RESPONSE: Thank you. These comments are helpful; terminology around model coupling differs across disciplines. Whilst we have developed WHAM! using JULES outputs, we here present WHAM! as though it is a standalone ('isolated' *sensu* Robinson) model, and hence do not describe it as coupled. This is for two main reasons.

Firstly, we consider WHAM! as presented here to be structurally a standalone model, able to be parameterised using inputs from several different sources. This is demonstrated by the capacity to substitute JULES inputs for Earth Observation data (Supplementary A).

Secondly, there are two forms of future coupling envisaged – an offline integration of WHAM! and INFERNO outputs, and a full online integration. To adopt the Robinson terminology (Robinson Figure 2), we currently have structurally isolated models, whilst an offline integration of WHAM! with INFERNO (paper in submission!) would be a 'prescribed periodic coupling', and a full online integration would be a 'periodic feedback coupling'. Periodic because WHAM! runs at coarser timestep than a typical DGVM. Hence, we describe WHAM! as standalone here to avoid confusion with these developments.

We have tried to clarify our terminology in several places: Line 91, Line 99, Lines 185-186, Lines 352-355. We have added the Best and Clark references, thanks for spotting that.

- Table 3: All sources are missing from reference list except for Hurtt et al. (2020).

RESPONSE: Thanks. Now added.

- Lines 401-407: It seems good that WHAM!'s burned area is almost always less than the GFED5-minus-GFED4 difference, since not all of that difference will be due to managed fires. (Although some managed fires will have been captured in GFED4 already.)

RESPONSE: We broadly agree here. Though, we would note that we think there are good reasons to believe GFED5 crop fires algorithm still does not detect a substantial portion of crop fires. See lines 629-635 in the main text.

- Fig. 4: Interpretation of this figure seems limited to evaluating regional trends, so it might make more sense to just show the 1990-2014 difference rather than separate 1990 and 2014 maps. However, since some inputs to WHAM! (namely potential evapotranspiration and ecosystem NPP) can exhibit strong interannual variation, it might be more appropriate to map either (a) the difference between means of 5-year periods or (b) trends determined from a linear regression in each gridcell.

RESPONSE: Thanks, good point. We have revised this figure such that it shows the mean values between 1990-2014 and the change in burned fraction over this period. We think this gives a clearer sense of overall model outputs. Now Figure 7.

- Lines 439-441:

   - I don't think Fig. 6b is informative as to the clustering of unmanaged fires in the WUI; the spatial resolution is just too coarse.

RESPONSE: This is a fair comment. We have been more explicit about what the map is showing, rather than adding our own interpretation.

   - How would a DGVM help determine industrialized firefighting effort? Isn't that what WHAM! is supposed to do?

RESPONSE: Hopefully this is now clarified in the methods section (2.4), lines 352-355. WHAM! provides a dimensionless suppression intensity (0-1). How this influences numbers of fires and indeed fire spread will need to be interpreted during the development of a model coupling, according to the structure and ontologies of the associated DGVM. As noted in the revised text, the logical impact in JULES-INFERNO would be a linear decline in numbers of unmanaged fires.

- It would be helpful to briefly mention what is needed from a DGVM (could use JULES-INFERNO as an example) to translate from number of fires to burned area.

RESPONSE: This has been added as supplement F.

- Fig. 7: Same comments as for Fig. 4.

RESPONSE: Thanks, we have amended figure 10 (formerly figure 7) in the same way as figure 7 (formerly figure 4).

- Lines 460-461 mention global trends and refer to Fig. 9, but that figure has no "Global" panel.

RESPONSE: We have added a global panel. (Now Figure 5).

- Fig. 10b: The caption should probably read "difference" instead of "change," the latter of which implies a change over time—especially when juxtaposed with 10a which does show change over time.

RESPONSE: We have amended as suggested.

- Lines 496-498: Asia and South America don't seem to have any dots around HDI 0.6.
RESPONSE: Thanks, this was a typo, corrected to 0.7.

- I think it would be easier on the reader to have "performance evaluation" type results (lines 401-407/Fig. 3b, Sect. 3.2) in a single section. This should probably be the first section of the Results, before scientific evaluation begins. So, e.g., Sect. 3.1.1 would be comparison against GFED4-5 difference, Sect. 3.1.2 would be comparison against GFED5 crop burned area.

RESPONSE: Agreed. We have re-ordered the results section.

- It would be really helpful to have figures in the Supplement illustrating various parts of the model. E.g.:

   - Geographic distribution of AFRs, AFTs, and LUSs.

RESPONSE: We have added a full set of maps of AFTs and LUSs to the Supplement (Supplement E). However, the AFRs were already published in Perkins et al., (2022) – hence we have not duplicated them here.

- It would be great to have a section on characteristics of DGVMs that are amenable to use of WHAM! E.g., "written in Python/able to call out to Python" or "calculate fire annually".

RESPONSE: We have added a supplementary F that addresses this. It's worth noting that, to our knowledge, no *online* ABM-DGVM coupling has yet been implemented, and so some points are not DGVM-specific but rather structural. For temporal alignment, e.g., Arneth et al., (2014) assume that an ABM will be running annually with a DGVM running at shorter timesteps, and so tackling this is going to be a general issue in coupled human-Earth system modelling rather than a WHAM! or DGVM-specific one. We would also note that much progress has been made in calling python libraries from Fortran (e.g.), and we are confident such issues can be solved similarly for WHAM!.

- Higher-resolution figures would be nice, as the text in e.g. Fig. 3 looks funky.

RESPONSE: Thanks – these are 300DPI. We can increase these for the final version if not clear.

Typos etc.:

- Line 364: "Teckentrup et al, (2018) pearson's" should be "Teckentrup et al. (2018), Pearson's"

- Line 429 (Fig. 5 caption): Colon is red for some reason.

- Line 566: Comma should be deleted.
- Supplement A, Sect. 1: "...firstly to WHAM! more readily transferrable..."

RESPONSE: These typos have been corrected.